# Drying Characteristics of a Combined Drying System of Low-Pressure Superheated Steam and Heat Pump

**Yuan Yao** [1,2,3], **Zhenneng Lu** [1,2,3,*], **Yulie Gong** [1,2,3], **Song Guo** [4,5], **Chupeng Xiao** [4,5] **and Wenbo Hu** [4,5]

1   Guangzhou Institute of Energy Conversion, Chinese Academy of Sciences, Guangzhou 510640, China; yaoyuan@ms.giec.ac.cn (Y.Y.); gongyl@ms.giec.ac.cn (Y.G.)
2   Key Laboratory of Renewable Energy, Chinese Academy of Sciences, Guangzhou 510640, China
3   Guangdong Provincial Key Laboratory of New and Renewable Energy Research and Development, Guangzhou 510640, China
4   State Grid Electric Power Research Institute Wuhan Efficiency Evaluation Company Limited, Wuhan 430070, China; guosong@sgepri.sgcc.com.cn (S.G.); xiaochupeng@sgepri.sgcc.com.cn (C.X.); huwenbo@sgepri.sgcc.com.cn (W.H.)
5   State Grid Electric Power Research Institute, Nanjing 211000, China
*   Correspondence: luzn@ms.giec.ac.cn; Tel.: +86-020-8705-8428

**Abstract:** The present study aimed at investigating the use of a drying system combining the concept of low-pressure superheated steam drying and heat pump drying for fish. The effects of various drying medium pressures on the temperature field, airflow field, drying time, equipment performance as well as the power consumption of the drying process were investigated and discussed. Four comparative tests with different initial pressures were carried out according to a specified drying process by the combined drying system. The results showed that when the vacuum was high, the temperature field and airflow field in the drying chamber were more uniform. Due to the poor heat transfer performance of the drying medium at high vacuum, the drying time increased with a decrease in initial pressure. It was also found that with the decrease in drying medium pressure, the power consumption of the heat pump and the axial fans was reduced, while the power consumption of the electric heater went up. Overall, the total power consumption is directly proportional to the drying medium pressure.

**Keywords:** combined drying; heat pump; low pressure; power consumption; drying time; heat transfer

## 1. Introduction

Due to oxygen-free drying medium, low temperature and high water evaporation rate, low-pressure superheated steam drying (LPSSD) has been proposed as an alternative method to dry foods, such as potato [1,2], pepper [3] and so on [4–6]. If the drying medium has normal pressure or high pressure, when the food surface temperature rises to the steam saturation temperature, melting, glass transition or damage will occur. Therefore, thermosensitive materials dried by LPSSD have superior quality than those dried by conventional drying. Chalida et al. [7] conducted a comparative study on bioactive compounds and bioactivities of centella asiatica by using different drying methods. The results showed that phenolic compounds, total triterpene saponins, antioxidant activity and antibacterial activity were the highest in the samples dried by LPSSD. Sehrawat et al. [8] evaluated the quality of dried mango blocks and onion slices by three drying methods: LPSSD, vacuum drying (VD) and hot air drying (HAD). The results showed that LPSSD resulted in the highest levels of ascorbic acid, carotene, total phenol and antioxidant activity. LPSSD of onion slices below 70 °C is the best drying condition, by which taste irritation, antioxidant activity, color and rehydration of onion slices are better than those dried by VD and HAD under the same conditions. Jamradloedluk et al. [9] and Phungamngoen et al. [10] compared the effects of HAD, VD and LPSSD on the rehydration of durian tablets and the

surface color of Chinese cabbage, respectively. The results showed that the rehydration of durian tablets dried by LPSSD was the best and the total color difference of Chinese cabbage dried by LPSSD changed the least.

By virtue of the low pressure, the saturated temperature of steam decreases and water evaporates at a lower temperature, which can not only maintain the quality of dried food, but also reduce the mass transfer resistance of water from food to a drying medium. The mathematical models of LPSSD under some applications have been preliminarily established and optimized. Elustondo et al. [11] proposed a semi-empirical mathematical model, based on a theoretical drying mechanism, and assumed that the water removal is carried out by evaporation in a moving boundary, allowing the vapor to flow through the dry layer built as drying proceeds. Defo et al. [12] developed a two-dimensional mathematical model based on the combination of the mass conservation equation and Darcy's law with negligible temperature gradients to simulate superheated steam vacuum drying of sugar maple sapwood. Suvarnakuta et al. [13] developed a simple three-dimensional liquid-diffusion model for LPSSD of a biomaterial. Sommart et al. [14] improved the model by considering the effect of initial steam condensation. The more accurate predictability of the optimized model was proven by the available experimental data [15]. Although LPSSD has some advantages in food drying, it does not always improve the drying efficiency compared with other drying methods, such as hot air drying (HAD), vacuum drying (VD) and heat pump drying (HPD). Some research results showed that in some drying processes, when the drying temperature is lower than a certain value, the drying time of LPSSD is longer than that of other drying methods. Leeratanarak et al. [16] found that when the drying temperature was below 80 °C, the potato chip drying time of LPSSD was longer than that of HAD. Devahastin et al. [16] and Panyawong et al. [17] studied LPSSD for heat-sensitive material using experimental methods. They also indicated that even when the drying temperature rose to 90 °C, the drying time of LPSSD was longer than that of VD. With the increase in drying temperature, the drying time difference between LPSSD and VD decreased.

Therefore, in order to improve drying efficiency and reduce drying energy consumption, some researchers combine LPSSD with other drying modes. Chatchai Nimmol et al. [18] carried out a test of drying banana slices with LPSSD and infrared radiation drying (IRD). During the test, the pressure of superheated steam was maintained at 200 kPa. The test results showed that when the drying temperature was greater than 90 °C, the drying rate of LPSSD combined with IRD is higher than that of IRD. Li et al. [19] studied design aspects, energetic performances and mathematical modeling on superheated steam drying. It was concluded that despite the fact that the drying time of LPSSD was significantly longer, the combined drying method of LPSSD and HAD would shorten the drying time and reduce the energy consumption compared with single LPSSD.

The above research results showed that under suitable drying conditions, the drying system combined with LPSSD and other drying methods can work at a higher drying rate. To attain the advantages of combined drying, a combination of LPSSD and HPD is proposed in this study. The effects of various drying medium pressure on the temperature field, airflow field, drying time, equipment performance as well as the power consumption of the drying process were experimentally investigated and discussed.

## 2. Materials and Methods

### 2.1. Experimental Set-Up

A combined drying system is proposed in this study, which can carry out low-pressure superheated steam drying and heat pump drying. Its schematic diagram is shown in Figure 1. Figure 2 shows the assembled drying system in the laboratory.

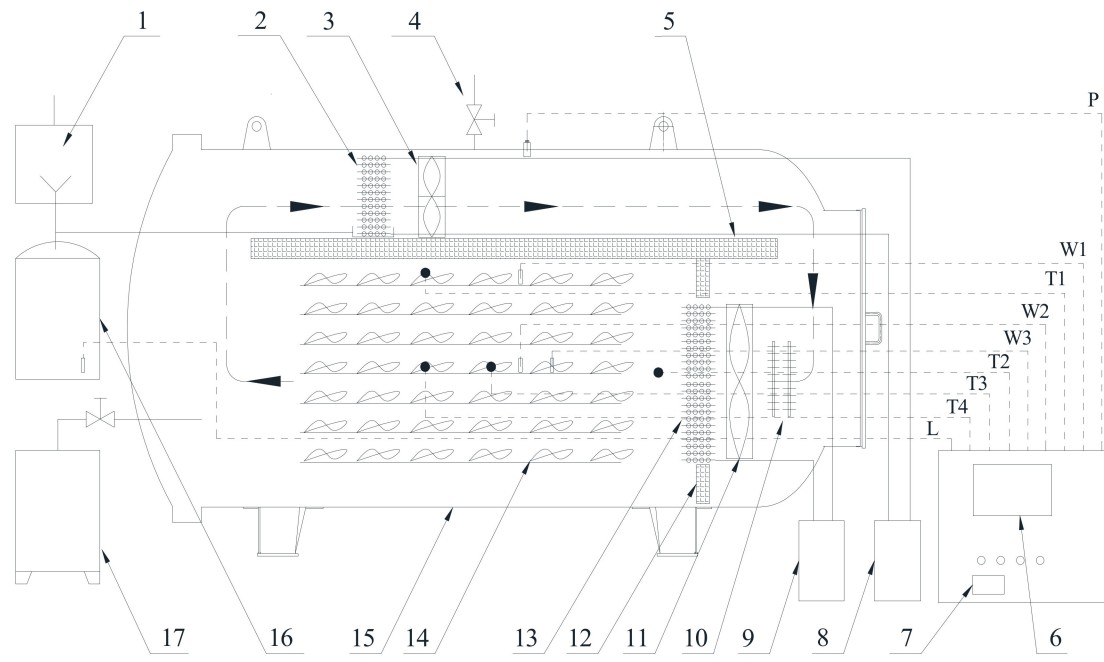

**Figure 1.** Schematic diagram of the drying system, with combined LPSSD and HPD. 1. Vacuum pump; 2. evaporator; 3. circulation fan; 4. intake valve; 5. insulated ceiling; 6. data recorder; 7. multi-load power meter; 8. dehumidification heat pump; 9. heating heat pump; 10. electric heater; 11. heating fan; 12. baffle; 13. condenser; 14. shelf; 15. vacuum chamber; 16. condensate storage tank; 17. steam generator.

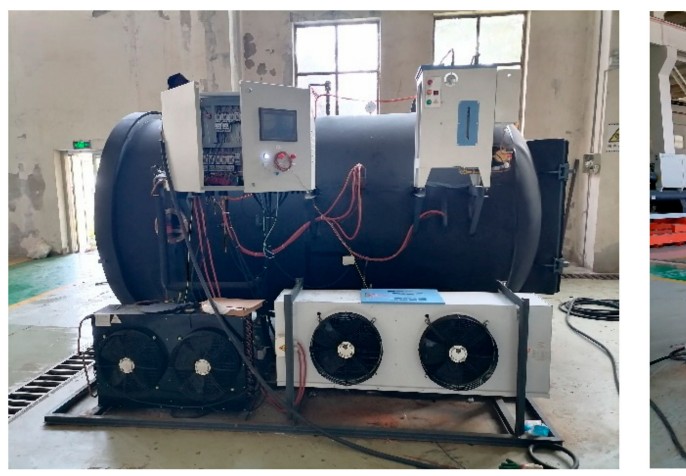

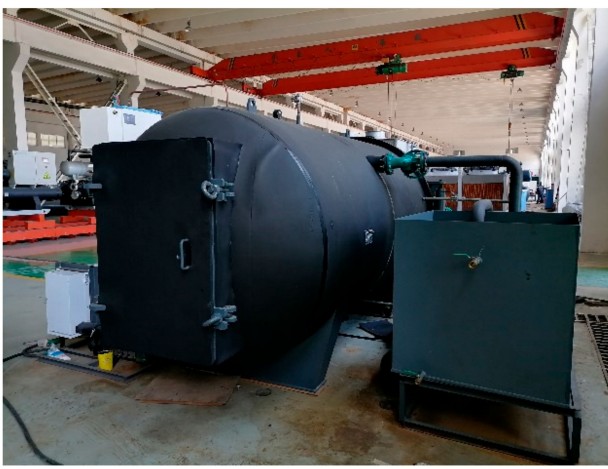

(**a**) the right side　　　　　　　　　　　(**b**) the left side

**Figure 2.** The assembled drying system with combined LPSSD and HPD.

The dryer mainly consists of a stainless-steel vacuum chamber (15), insulated with rubber insulation material, with inner dimensions of $\Phi 182 \times 400$ cm$^3$; a steam generator (17) rated 3 KW, which produced the steam from a boiler; a water ring vacuum pump (1) rated at 7.5 KW, which was used to create a vacuum in the drying chamber; a dehumidification heat pump (8) rated at 2.2 KW, in which an evaporator (2) can condense vapor from drying material; a heating heat pump (9) also rated at 2.2 KW, in which a condenser (13) can heat the drying medium; an electric heater (10) rated at 5 KW, which works when the heating heat pump stops working; two axial fans, respectively, rated at 0.74 KW and 1.5 KW (3, 11), which are used to circulate the drying medium in the chamber; and a data recorder (6), which can not only record test data, but also display the real-time data.

In addition to the above main functional components, some test instruments are also installed. A pressure sensor (P) and a thermocouple (T2) are used to test the pressure and temperature of the drying medium, respectively. A multi-load power meter (7) is assembled in the data recorder to measure the power consumption of each electric device in the dryer. Three thermocouples (T1, T3, T4) and three air speed sensors (W1, W2, W3) are used for testing the temperature of drying fishes and wind speed in different positions (see Figure 3 for details). A liquid-level sensor (L) is installed on the condensate storage tank (16) to measure the amount of evaporated water, so that the mass and moisture content of the drying material can be calculated in real time. Details of all the above-mentioned test instruments are shown in Table 1.

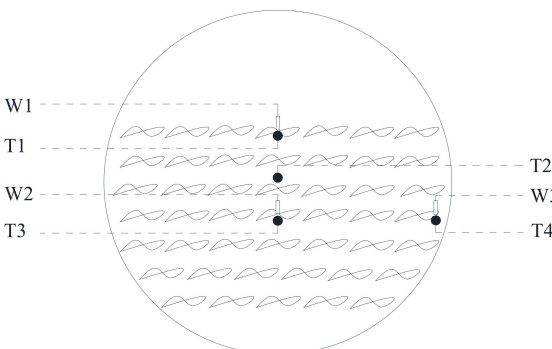

**Figure 3.** Distribution diagram of temperature and air speed test points.

**Table 1.** Details of the test instruments in the combined drying system.

| Instruments | Manufacturer and Type | Measuring Range | Measurement Accuracy |
|---|---|---|---|
| Pressure sensor (P) | Asmik, MIK-P300 | −100 Kpa~0 Pa | 0.5% |
| Thermocouple (T1, T2, T3, T4) | OMEGA, TT-K-30-SLE | −270~400 °C | 0~400 °C, 0.75% −270~0 °C, 2% |
| Power meter (7) | Schneider, DM2350 N | | 0.5% |
| Air speed sensors (W1, W2, W3) | Huayu, WD400 | 0~10 m/s | 3% |
| liquid level sensor (L) | ELECALL, ELE-802 | 0.01~5 m | 0.5% |

### 2.2. Material

Dried fish is a popular food in China. Improving the quality of dried fish and reducing power consumption of the dryer has always been the goal of the dried fish producers. The combined drying of LPSSD and HPD is a potential method to achieve this goal. In this study, grass carp was chosen as the drying material. Fresh grass carp has an initial moisture content (wet basis) in the range of 70~75% [20]. The test fish were purchased from a local supermarket. Before being sent into the drying chamber they were gutted and scaled. The total weight of these eviscerated carp was 50 kg. For the sake of same drying rate, the weight of each test fish was about 1.5 kg. Figure 4a,b show the test fish before drying and dried fish, respectively.

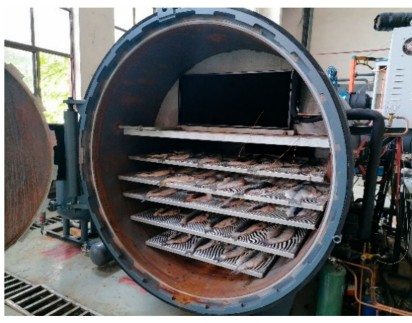
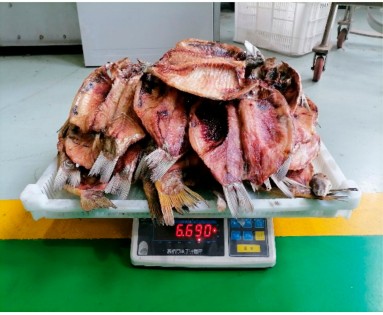

(**a**) fish before drying          (**b**) dried fish

**Figure 4.** The fish in the drying tests.

### 2.3. Methods

The methods for the combined drying using this experimental set-up contain the following steps:

- Preheat

To reduce the amount of steam condensation in the drying chamber during the start-up of tests the drying chamber must be preheated. The heating heat pump and the two axial fans were turned on to heat up the drying chamber until their temperatures reached the desired value without the application of steam to the drying chamber during the initial stage of the drying process.

- Create vacuum (or low pressure)

Firstly, the prepared fresh grass carp (gutted and scaled) were placed on the shelf evenly. Secondly, the seal door of the drying chamber was closed tightly. Then the vacuum pump was switched on to exhaust air in the drying chamber until the pressure dropped to the preset value.

- Produce superheated steam

The steam generator was turned up to supply saturated steam to the drying chamber. When the saturated steam entered the vacuum chamber, it became superheated steam due to the low pressure. To increase the superheat, the heating heat pump or electric heater could be turned on if necessary.

After the above three steps were completed, the drying fish tests of combined LPSSD and HPD began to be carried out stably. The temperature of superheated steam was controlled by heating heat pump (or electric heater) and dehumidification heat pump. The pressure in the drying chamber was determined by the amount of reserved air and the steam evaporated from the fish. The vacuum pump and the steam generator only worked at the initial stage of the tests. When the desired pressure was reached, they were turned off. In the whole test process, the two axial fans always worked. The drying temperature and pressure were adjusted only by stopping and turning on the heating heat pump and dehumidification heat pump. When the heating heat pump stops working because of overheating protection, the electric heater starts as an auxiliary heat source.

### 2.4. Drying Process and Operating Parameters

Drying process and operating parameters are very important to dried food. Comprehensively considering drying rate, moisture content, nutrition, taste and other factors of dried foods, could lead to the right drying process with appropriate parameters, only after many tests and improvements. The drying process and parameters of grass carp in this test (see Table 2 for details) were determined according to the dried aquatic product standard (NY/T 1712-2018) issued by China's Ministry of Agriculture and some related literature [21–23].

**Table 2.** Drying process and parameters of dried grass carp.

| Stage | Drying Medium Temperature(°C) | Moisture Content (kg/kg ×100%, (w.b)) |
|-------|-------------------------------|----------------------------------------|
| 1 | 27~32 | 70~50% |
| 2 | 32~37 | 50~40% |
| 3 | 37~42 | 40~30% |
| 4 | 42~47 | 30~20% |

As mentioned previously, drying efficiency of LPSSD is lower than that of hot air drying, especially at low temperatures. This is mainly because the density of low-pressure superheated steam is much lower than that of air, which results in low enthalpy per unit volume and low convective heat transfer coefficient (HTC). Peamsuk [24] calculated that the HTC of LPSSD was less than 10 W/($m^2 \cdot$k) when the drying temperature was 60~80 °C and the operating pressure was 7~13 KPa. For this reason, the drying media of these tests are, respectively, composed of superheated steam and air in different proportions.

The initial pressure represents the amount of air reserved in the drying chamber after vacuum pumping. Four tests were carried out in this study, and their initial pressure and composition are listed in Table 3.

**Table 3.** Composition of the initial drying medium in tests.

| Test No. | Initial Pressure (KPa) | The Amount of Air Reserved | The Amount of Steam Inputed (kg) |
|---|---|---|---|
| 1 | 7 | about 7% air reserved | 1 |
| 2 | 35 | about 35% air reserved | 1 |
| 3 | 65 | about 65% air reserved | 1 |
| 4 | 101 | about 100% air reserved | 0 |

## 3. Results and Discussion

### 3.1. Temperature Field Characteristics

As shown in Figure 3, T2, T1, T3 and T4 represent the temperature of the drying medium, the temperature of the fish placed at the middle-upper location, the temperature of the fish placed at the central location and the temperature of the fish placed at the edge location, respectively. As we know, the temperature field in the drying chamber is uneven. The effect of the drying medium with different initial pressures on the uniformity of the temperature field in the drying chamber is studied in this section. Figure 5 shows the temperature difference between T2 and T3, T2 and T1, T2 and T4 under different initial pressures during the whole drying process. As can be seen from Figure 5, when the initial pressure is smaller, the difference in the values of temperature difference between T2 and other temperatures is relatively close. This shows that when the vacuum is high, the temperature field in the drying room is relatively uniform, which is conducive to the consistency of dried products. However, the higher temperature difference indicates that when the vacuum degree is high, the heat transfer between the drying medium and the drying material is poor. It should be noted that the big temperature difference at the beginning of drying is due to the fact that the fish were taken out from the freezer before they were placed on the shelf.

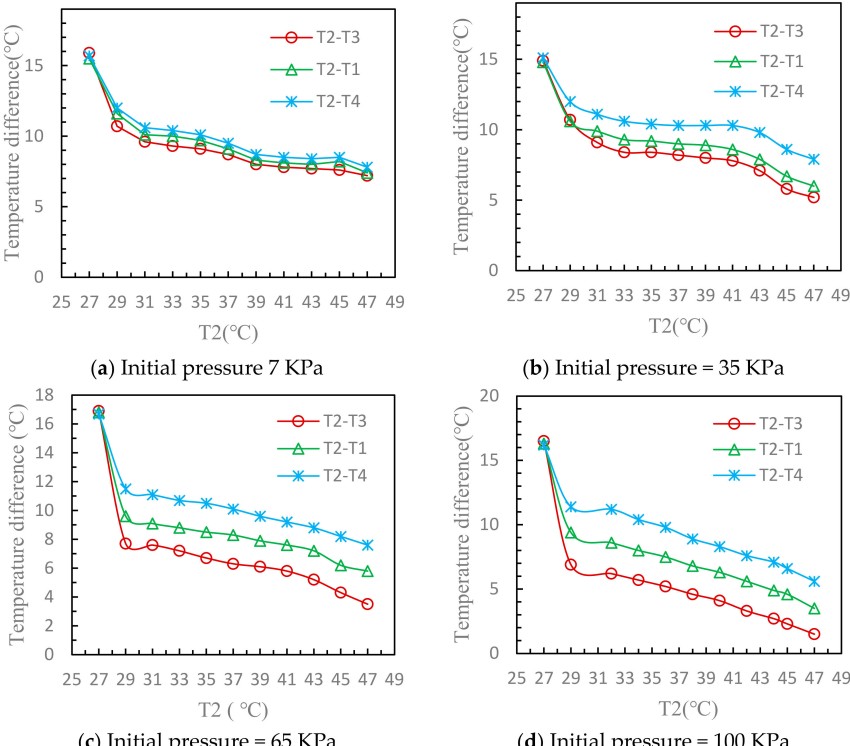

**Figure 5.** Variation in temperature difference during drying under different initial pressures.

### 3.2. Wind Speed Field Characteristics

Figure 6 shows the corresponding changes in wind speed and drying medium pressure under different initial pressures in the specified drying process. Firstly, it is seen from Figure 6 that the variation range in drying medium pressure decreases with the increase in initial pressure. When the initial pressure is 7 KPa, 35 KPa, 65 KPa and 101 KPa, respectively, the difference between maximum pressure and minimum pressure of drying medium is 13.1 KPa (19.8~6.7), 8.6 KPa (43.5~34.9), 7 KPa (72.9~65.9) and 6.1 KPa (101.3~95.2), accordingly. This is because water evaporates more easily when the pressure is lower. The content of water vapor determines drying medium pressure during drying. Secondly, wind speed is significantly affected by initial pressure. As revealed by these figures the greater the pressure, the greater the wind speed, since high initial pressure actually represents more air in the drying medium, which has greater density. When the fan speed remains constant, wind speed increased with the accretion of drying medium density. Thirdly, it is found that the difference in wind speed at each test point (W1, W2, W3) also increases with an increase in initial pressure. This result shows that a lower pressure will produce a more uniform air flow field, which is conducive to the drying consistency of dried products.

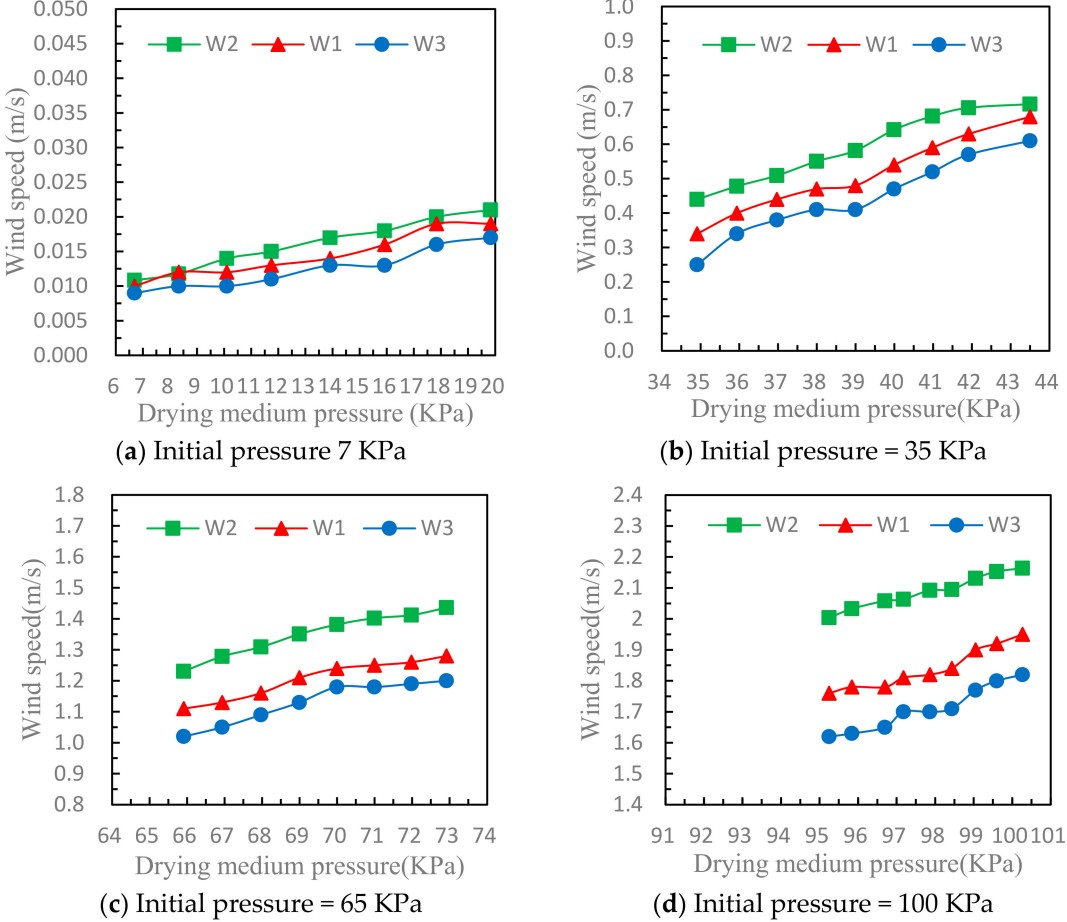

**Figure 6.** Variation in wind speed during drying under different initial pressures.

### 3.3. Drying Time

It is well known that drying time reflects drying efficiency. Figure 7 shows the drying curves for the test fish in the whole drying process undergoing different initial pressures. It can be seen from these curves that the drying time increases with a decrease in initial pressure. This phenomenon is in agreement with that reported by Leeratanarak et al. [16] and Li et al. [25], who found the drying time of LPSSD is longer than that of HAD when drying is conducted at lower temperatures. This is because the temperature difference

between the test fish and drying medium at a lower drying temperature was less than that at a higher temperature, hence, a smaller driving force for heat transfer, which is also related to the rate of mass transfer. Although the lower pressure will make it easier for water to evaporate from the fish surface to the drying medium, the low heat and mass transfer efficiency caused by low pressure make it difficult for water in the fish to migrate from the inside to the surface. It is also found that the moisture ratio difference between high pressure and low pressure decreases in the later stage of drying. Therefore, it can be predicted that the drying time of LPSSD will be less than that of HAD in this test when the drying temperature rises to a certain value, which is called inversion temperature [26].

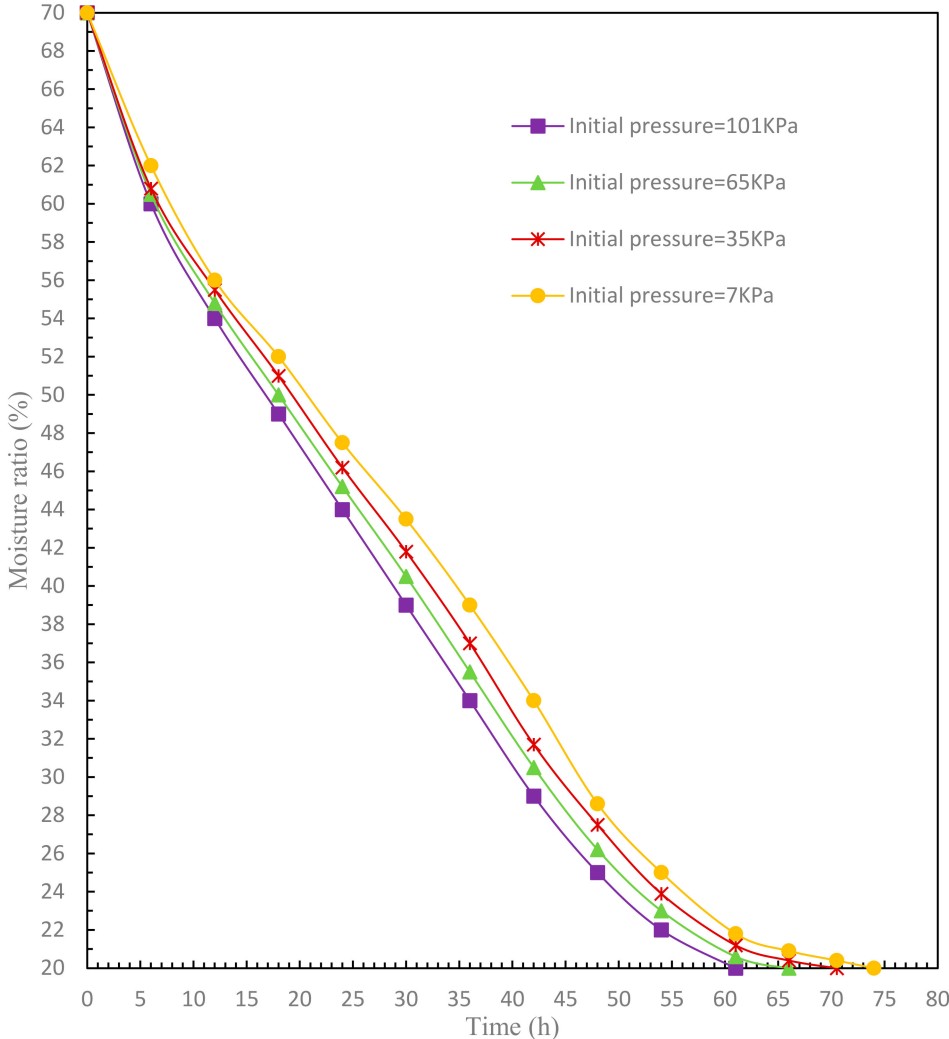

**Figure 7.** Drying time under different initial pressures.

### 3.4. Device Working and Energy Consumption

As described in Section 2.1, several devices are installed on the experimental set-up, which can work as a combined drying system. For engineering applications, in addition to the drying characteristics, the working performance and energy consumption of this equipment under different drying medium pressures must also be studied. Figure 8 shows the total power consumption in different tests. From the results shown in the curves, although the drying time of test No. 1 (initial pressure = 7 KPa) is the longest, its power consumption is the least. As for the other tests, the total power consumption from low to high is successively Test No. 2 (initial pressure = 35 KPa), Test No. 3 (initial pressure = 65 KPa) and Test No. 4 (initial pressure = 101 KPa). The reason why the power consumption is inversely related to the drying time is that the working performance of each piece of electrical equipment in the

combined drying system varies widely under different pressures. The power consumptions of the electrical equipment in the four tests with different initial drying medium pressures are shown in Figure 9.

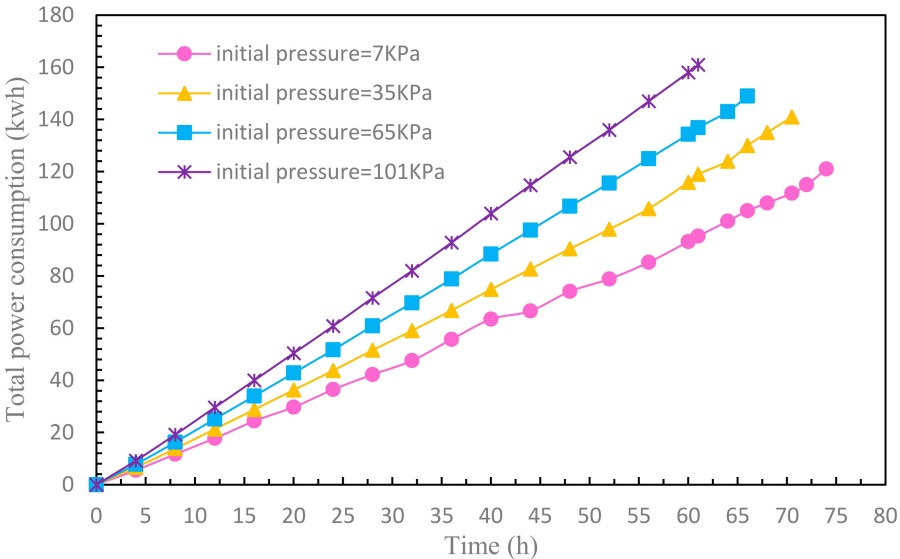

**Figure 8.** Total power consumption under different initial pressures.

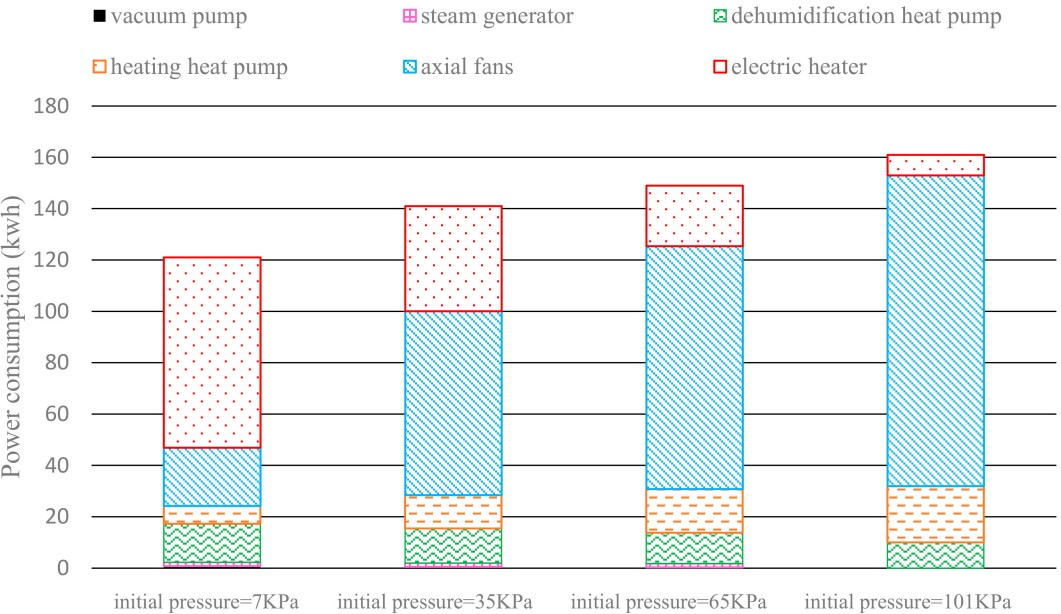

**Figure 9.** Power consumptions of electrical equipment under different initial pressures.

The experimental device has two heat pumps; one is a dehumidification heat pump, the other is a heating heat pump. In all the tests, the dehumidification heat pump worked normally, while overload protection often occurred on the heating heat pumps in low-pressure tests, especially in Test No. 1. This is due to the bad heat transfer of the low-pressure drying medium, which makes the condenser unable to diffuse a large amount of heat generated by the compressor in time, resulting in the suspension of the compressor because of overheating [27]. It is seen from Figure 9 that the power consumption of the heating heat pump was reduced with the decrease in initial pressure of the drying medium.

Besides the heating heat pump, the power consumption of the axial fans also changes significantly with the drying medium pressure. According to fan law, the power of the axial fan is directly proportional to air pressure and air volume [28]. Lower air pressure

reduces the resistance of the fan blades to rotation. The output power of the fan motor is reduced accordingly. Consequently, it is found from Figure 9 that the power of the axial fan decreases with the pressure in the drying medium. This is the main reason why Test No. 1 consumed less power than the other tests, even if its drying time is the longest.

Compared with the heat pump, the heating energy efficiency ratio of the electric heater is much lower. Therefore, according to this drying process control strategy, the electric heater only works when the heating heat pump stops due to overheating protection. With a rise in drying medium pressure, the heat dissipation in the heat pump compressor was gradually promoted. Therefore, the working time of the heating heat pump became longer, while the working time of the electric heater became shorter. As a result, it is seen from Figure 9 that the power consumption of the electric heater descended as the drying medium pressure ascended.

Since the vacuum pump and the steam generator only worked for a short time in the initial stage of drying, their proportion in the total energy consumption is very low. Even in test No. 4, they did not work at all. Therefore, the change in energy consumption of the vacuum pump and the steam generator in the drying process was ignored and will not be discussed in detail.

## 4. Conclusions

A drying system combining the concept of low-pressure superheated steam and heat pump for drying fish was developed and studied. In the specified drying process, four tests were conducted with different initial pressures of drying medium. The results showed that when the vacuum was high, the temperature field and airflow field in the drying chamber were more uniform, which was conducive to the consistency of dried fish. However, the high temperature difference between the drying medium and the drying material indicated that the heat transfer was poor undergoing the high vacuum. Therefore, it was found that the drying time increased with a decrease in initial pressure. Due to the increasing drying temperature, the moisture ratio difference between high pressure and low pressure decreased in the later stage of drying.

The application of a heat pump in low-pressure superheat drying is a great challenge to the stability of the heat pump at high temperature. When the pressure in the drying medium decreases, the number of shutdowns in the heating heat pump because of overheating protection increases. Owing to the decrease in static pressure, the input power of the axial fans in LPSSD decreases greatly. For the above reasons, with the decrease in drying medium pressure, the power consumption of the heating heat pump and the axial fans will be reduced, while the power consumption of the electric heater will go up. Overall, the total power consumption is directly proportional to the drying medium pressure.

**Author Contributions:** Conceptualization, Y.Y.; methodology, Z.L.; investigation, Y.Y. and Z.L.; resources, S.G. and C.X.; project administration, Y.G.; validation, W.H.; writing—original draft preparation, Y.Y.; writing—review and editing, Z.L. and Y.G. All authors have read and agreed to the published version of the manuscript.

**Funding:** This research was funded by the Science and Technology Projects of State Grid Corporation of China, grant number 5400-202140401A-0-0-00.

**Institutional Review Board Statement:** Not applicable.

**Informed Consent Statement:** Not applicable.

**Data Availability Statement:** Not applicable.

**Acknowledgments:** Thanks to others in the author's research group for their support in this work.

**Conflicts of Interest:** The authors declare that there is no conflict of interest.

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
