# Peer review of "Drying Characteristics of a Combined Drying System of Low-Pressure Superheated Steam and Heat Pump"

_processes, doi:10.3390/pr10071402_

Round 1

Reviewer 1 Report

The article concerns very interesting experimental studies of the fish drying system through the use of the low-pressure concept of superheated steam and a heat pump. The influence of drying agent pressure on the temperature field, air flow field, drying time, equipment efficiency and energy consumption of the drying process were analyzed. I consider the reviewed article to be prepared at the appropriate substantive level for publication in the Processes journal. Nevertheless, I have a few remarks that require corrections, comments or clarifications. 1. Chapter 0 is to be removed, 2. In my opinion, the introduction would need to be extended by including more bibliographic items, 3. The work is experimental, so in my opinion it would require a broader description of the measuring devices, error analysis, as well as the method of planning or conducting the experiment, 4. The summary does not provide any relevant final conclusions, apart from a simple description of the results obtained. In my opinion, on the basis of the results obtained, guidelines should be indicated that allow for the selection of steam pressure in the fish drying system due to some criteria. I missed that in this fragment, 5. The bibliography containing 22 seems to be too poor and in my opinion needs to be supplemented. 

Reviewer 2 Report

The article presents the performance of a combined low pressure steam and heat pump drying process for the food industry. The article must be improved as it appears more as a technical report than as a research article. The literature overview must be expanded, the scientific contribution must be better stated, and the results and discussion section must be improved with better explanations of the physical phenomena behind low pressure drying as well as with the comparisons against previous research. In this regard, the authors may find the following comments helpful:

  1. The instructions to authors in lines 29-35 and 169-171 must be deleted.
  2. The literature references should appear normally inside the text such as [1], [2-5], and not in the superscript position [1], [2-5].
  3. The literature overview, concerning the latest findings on LPSSD (lines 45-65) must be expanded in order to better understand the problems in the previous research and the proposed scientific advancement of the present study. What are the most recent findings on LPSSD-HPD combined drying and how will the present research expand this knowledge?
  4. Line 112: the weight of 50 kg is before or after removing the fish guts and scales?
  5. Provide more details about the measurement equipment: type of sensors, uncertainty, number of measurements, time period for data collection, etc.
  6. Table 2: the mass of steam inside the drying chamber was always 1 kg, for all vacuum pressures?
  7. Line 161: the HTC of 10 W/m2K is for forced or natural convection inside the drying chamber? What is the usual air/steam velocity inside the drying chamber?
  8. The figure numbering is incorrect: two figures 2 then comes figure 5.
  9. Table 2 reports the initial pressures - how does this pressure change over time? Beside the pressure increase caused by moisture evaporation, is there air infiltration from the outside to the chamber? Why is the vacuum pump not started to maintain the pressure inside the chamber constant and equal to the initial pressure?
  10. Figure 5 shows the temperature differences when T2 is the referent temperature for comparison with the other temperatures T1, T3 and T4. Why is the temperature T2 always the highest temperature inside the chamber?
  11. Line 186 says that the big temperature difference at the beginning of drying is caused by taking the cold fish out from the freezer, but line 124 says that the fish is preheated using the heat pump before the beginning of the drying process.
  12. Lines 240-242: the explanation about why the power consumption is inversely proportional to the drying pressure is vague - it must be improved. This inverse relation between power consumption and drying pressure can be backed up with literature references?
  13. The reduced power consumption at low drying pressures is a result of the improved moisture transfer between fish and air or it is also caused by the adverse effects of low pressure on the power consumers?
  14. The low pressure inside the drying chamber could cause increased maintenance and repairs to the equipment, which would increase the drying costs?
  15. Figure 7 shows that the necessary drying time is more than 70 hours (about 3 days) to achieve moisture content of 20%. This drying time for fish can be compared with literature references for different drying methods (HAD, VD, IRD, LPSSD, combined processes)?

Reviewer 3 Report

Dear Auhtors,

you paper can be accepted to the print after providing minor changes:

  • pease remove "0" and other template text: "0. How to Use This Template"
  • Introduction section has poor quality, and need to be improved and extended of review the research of other researchers in presented topic/topics
  • In chapter 3.2, I propose to use "air speed" instead of "wind speed"

Round 2

Reviewer 2 Report

After the first round of revisions, the authors have addressed my comments and improved the manuscript. I still have several minor suggestions:

1. The total weight of the fish is 50 kg and it refers to the weight after removing guts and scales - this should be clearly stated in the article. In the article it is not clear if the weight of 50 kg refers to before or after removing guts and scales.

2. The drying steps are described in section 2.3. In the first step you mention that the chamber is preheated using the heat pump. What temperature is achieved and what is the time interval of the preheating process? How much moisture was removed from the fish during the preheating step? In the second step the fresh carp is placed on shelves at intervals. This means that you slowly loaded the chamber with the fish, the total mass of 50 kg of fish was not put inside all at once? What were the intervals for the loading process?

3. What was the pressure inside the drying chamber during the 4 stages of table 2? The 4 stages of table 2 are the 4 steps of section 2.3 (preheating, vacuum generation, superheated steam production, drying)? Table 2 gives the initial pressure inside the drying chamber - this pressure is constant or variable during the entire drying process?

4. The instruction to authors (lines 204-206) must be deleted.

5. You mention that the main purpose of preheating is to prevent steam from condensing on the fish surface in the beginning. What would happen to the drying efficiency if the steam had condensed on the fish surface (without preheating stage)?

6. The Abstract mentions that the total power consumption is inversely related to the drying pressure (lines 29-30). However, figure 8 shows that the total power consumption decreases with the reduction of drying pressure, which means that the relation is directly proportional?

7. The reduced power consumption at lower drying pressures is mainly a consequence of the decreases power consumption of axial fans (figure 9). What would happen to the drying time and drying efficiency if the axial fans were completely switched off during the entire drying process? The conclusion about the reduced total power consumption at lower drying pressure would not be true anymore?
